# Response of *Chaetomium* sp. to Nitrogen Input and Its Potential Role in Rhizosphere Enrichment of *Lycium barbarum*

**DOI:** 10.3390/microorganisms13081864

**Published:** 2025-08-09

**Authors:** Ru Wan, Hezhen Wang, Xiaojie Liang, Xuan Zhou, Yajun Wang, Yehan Tian, Zhigang Shi, Yuekun Li

**Affiliations:** 1National Wolfberry Engineering Research Center, Wolfberry Science Research Institute, Ningxia Academy of Agriculture and Forestry Sciences, Yinchuan 750002, China; wanru2008@163.com (R.W.); lxj910303@126.com (X.L.); m18695145633@163.com (X.Z.); yajun817@163.com (Y.W.); 2College of Landscape Architecture, Beijing Forestry University, Beijing 100083, China; hezi184@bjfu.edu.cn; 3College of Plant Protection, Shandong Agricultural University, Taian 271018, China; tianyehan@163.com

**Keywords:** goji berry, fungal community, beneficial fungi, growth promotion, nitrogen cycle

## Abstract

*Lycium barbarum* L. (goji berry), a traditional Chinese medicinal plant, depends heavily on nitrogen input to maintain productivity. Nitrogen application also profoundly influences rhizosphere microbial dynamics, which are critical for soil health and plant performance. This study aimed to investigate how the rhizosphere fungal community responds to nitrogen input and explore the potential role of beneficial fungi (e.g., *Chaetomium*) in goji berry rhizosphere enrichment. A field experiment with four nitrogen levels (0, 53.82, 67.62, and 80.73 g·N m^−2^·year^−1^, designated as N0, N1, N2, and N3) was conducted to analyze the fungal community structure in the rhizosphere of goji berry using ITS rRNA gene amplicon sequencing. The results showed that nitrogen input significantly altered the rhizosphere fungal community composition and diversity. Redundancy analysis (RDA) and Mantel tests indicated that soil electrical conductivity, total phosphorus, available phosphorus, and nitrate nitrogen were key environmental factors driving the fungal communities’ shifts. Notably, specific fungal genera, including *Chaetomium*, *Cladosporium*, *Gibberella*, *Fusarium*, *Pyxidiophora*, *Acaulium*, and *Lophotrichus*, exhibited differential enrichment across nitrogen levels. In particular, *Chaetomium* was significantly enriched under the conventional nitrogen treatment (N2), a strain of *Chaetomium* sp. LC101 was successfully isolated from the goji berry rhizosphere, and its functional roles were verified via pot experiments. Inoculation with *Chaetomium* sp. LC101 significantly promoted goji berry growth, with the most pronounced effects observed under N0 treatments, root fresh weight, root vitality, and leaf chlorophyll content increased by up to 55.10%, 15.69%, and 43.27%, respectively, compared to non-inoculated controls. Additionally, *Chaetomium* sp. LC101 regulated rhizosphere nitrogen transformation by enhancing urease, nitrite reductase, and polyphenol oxidase activities while inhibiting nitrate reductase activity. These findings demonstrate that *Chaetomium* responds sensitively to nitrogen input, with enrichment under moderate nitrogen levels, and acts as a beneficial rhizosphere fungus by promoting plant growth and regulating nitrogen cycling. This study provides novel insights for nitrogen management in the goji berry industry, where synergistic regulation via “nitrogen reduction combined with microbial inoculation” can reduce nitrogen loss, improve yield and quality through functional fungi, and contribute to ecological sustainability.

## 1. Introduction

Nitrogen, a core nutrient indispensable for plant growth, directly determines the productivity of agricultural ecosystems [1]. Urea, as the most widely used nitrogen fertilizer in agriculture, can increase crop yield and maintain soil health by regulating the functional dynamics of rhizosphere microbial communities when applied rationally [2]. However, in traditional agricultural systems, high crop yields rely heavily on high nitrogen fertilization, with a substantial portion of nitrogen lost by leaching rather than being absorbed by plants. In addition, the overapplication of nitrogen not only causes environmental pollution but also disrupts healthy microbial communities in the rhizosphere, destabilizes microbial networks, and significantly increases community vulnerability to environmental disturbances [3].

An imbalanced nitrogen input (excess or deficiency) disrupts the equilibrium of soil microbial communities, particularly altering the composition and functions of rhizosphere fungi. Long-term nitrogen application drives fungal community enrichment in Ascomycota and Basidiomycota by modifying soil pH, electrical conductivity, organic matter content, and root exudate profiles. Concurrently, the abundance, richness, and diversity of arbuscular mycorrhizal fungi (AMF) decrease with increasing nitrogen addition rates [4,5,6]. Nitrogen addition also induces seasonal shifts in the AMF community structure, with Glomus emerging as the dominant genus under high-nitrogen conditions, whereas Diversispora abundance decreases, highlighting a dose-dependent response of fungal communities to nitrogen input [7]. Notably, excessive nitrogen input may reduce functional redundancy, promote the replacement of oligotrophic fungi by eutrophic taxa, weaken the soil nitrogen retention capacity, and exacerbate the risk of leaching [8]. Conversely, reducing nitrogen application can promote the proliferation of nitrogen-fixing taxa (e.g., Mesorhizobium and Bradyrhizobium) and stabilize microbial networks [9]. These findings underscore the sensitivity of rhizosphere microbiomes to nitrogen management and highlight how excessive nitrogen input disrupts the delicate balance of nitrogen cycling and ecosystem health by favoring fast-growing eutrophic fungi over oligotrophic groups, reducing functional redundancy, and compromising soil nitrogen retention [10,11,12].

As an economically important crop with high medicinal and nutritional value, the stability of the rhizosphere microenvironment directly impacts the yield and quality of *Lycium barbarum* (L. (Solanaceae). Our previous research revealed that nitrogen fertilization levels are closely associated with the physicochemical properties (e.g., pH, electrical conductivity, and organic matter content) and enzyme activities (e.g., urease and nitrate reductase) of *L. barbarum* rhizosphere soil, indirectly regulating the dynamics of rhizosphere bacterial communities [1]. The present study further investigated the sensitivity of *L. barbarum* fungal communities to nitrogen input and explored the effects of functional fungi on plant growth, aiming to develop synergistic strategies for nitrogen management and microbial regulation (e.g., nitrogen reduction combined with microbial inoculation), providing new perspectives for the sustainable development of *L. barbarum* and the maintenance of ecosystem functions.

## 2. Materials and Methods

### 2.1. Site Description and Experimental Design

The N enrichment experiment was performed in the *Lycium barbarum* germplasm nursery of the Institute of Goji berry science of NingXia Academy of Agricultural and Forestry Sciences, NingXia, China (36°25′48″ N, 106°09′00″ E, 1428.5 masl.), with an average annual precipitation of 367.4 mm, a mean annual temperature of 7.3 °C, and an average annual sunshine duration of 2710 h. Four levels of nitrogen enrichment were used: 0 (blank, N0), 53.82 (nitrogen reduced 20%, N1), 67.62 (conventional nitrogen, N2), and 80.73 (nitrogen increased 20%, N3) g·N-m^−2^·year^−1^, and each treatment was repeated five times. Among these, N2 is the most commonly used fertilization dosage for the field production of goji berries. Seven-year-old *L. barbarum* plants from the Ningqi No. 7 cultivar were used as the experimental material. The fertilization regimen consisted of a single basal application followed by three supplemental top-dressings administered at key growth stages: the initial application on 28 April 2021, and subsequent applications on 29 May and 7 July 2021.

Rhizosphere soil samples were collected on 16 July 2021. Three to five healthy plants from the experimental units were randomly selected for each treatment, and the sampled trees were labeled. Rhizosphere soil samples from 0 to 40 cm vertical depths attached to the root cap were collected using the 5-point sampling method. About the same soil volume was collected at each sampling point. After mixing, a part of the soil sample was loaded into frozen storage tubes, stored in liquid nitrogen tanks, subsequently transferred to the laboratory, and stored at −80 °C for amplicon sequencing experiments. An additional portion of the soil sample was air-dried in the shade for the determination of soil chemical properties, as summarized in our previously reported study [1].

### 2.2. DNA Extraction, PCR Amplification, and Sequencing

Total microbial genomic DNA was extracted from rhizosphere soil samples using the E.Z.N.A.® Soil DNA Kit (Omega Bio-Tek, Norcross, GA, USA) in accordance with the manufacturer’s protocol. DNA quality and concentration were assessed using 1.0% agarose gel electrophoresis and a NanoDrop 2000 spectrophotometer (Thermo Scientific, Waltham, MA, USA), and the samples were stored at −80 °C for subsequent analysis. The hypervariable ITS1–ITS2 region of the fungal ITS rRNA gene was amplified using the primer pair ITS1F (5′-CTTGGTCATTTAGAGGAAGTAA-3′) and ITS2R (5′-GCTGCGTTCTTCATCGATGC-3′) with a T100 Thermal Cycler (Bio-Rad, Hercules, CA, USA). Each 20 µL PCR mixture contained 4 μL of 5× Fast Pfu buffer, 2 μL of 2.5 mM dNTPs, 0.8 μL of each primer (5 μM), 0.4 μL of Fast Pfu polymerase, 10 ng of template DNA, and nuclease-free water. The PCR amplification conditions were as follows: initial denaturation at 95 °C for 3 min; 27 cycles of denaturation at 95 °C for 30 s, annealing at 55 °C for 30 s, and extension at 72 °C for 45 s; a final extension at 72 °C for 10 min; and holding at 4 °C. The PCR products were separated using 2% agarose gel electrophoresis, purified via the PCR Clean-Up Kit (YuHua, Shanghai, China), and quantified using a Qubit 4.0 fluorometer (Thermo Fisher Scientific, USA).

### 2.3. Sequencing and Statistical Analyses

The raw sequencing data were demultiplexed via an in-house Perl script, followed by quality filtering with fastp version 0.19.6 and sequence merging using FLASH version 1.2.7 [13,14]. The optimized sequences were clustered into operational taxonomic units (OTUs) at a 97% sequence similarity threshold using UPARSE 11.0.667 [15,16]. The most abundant sequence within each OTU was selected as the representative sequence. To minimize the influence of sequencing depth on alpha and beta diversity assessments, the number of ITS rRNA gene sequences per sample was rarefied to 20,000, yielding an average Good’s coverage of 99.09%. Rarefaction analysis was performed using Mothur v.1.30.1 [17], whereas the vegan 2.4.2 package was employed to calculate alpha diversity indices and Good’s coverage [18,19,20]. Significant differences in alpha diversity among the four nitrogen levels were assessed using one-way analysis of variance (ANOVA).

Histograms illustrating the fungal community structure were generated using the R software package vegan 2.4.2 [18]. Principal coordinate analysis (PCoA) and redundancy analysis (RDA) were conducted to visualize the microbial community ordination. To assess microbial community similarity and the effects of different growth stages and spatial compartments on community composition, analysis of similarities (ANOSIM) and permutational multivariate analysis of variance (PERMANOVA) were performed. Biomarkers specific to four nitrogen levels were identified using linear discriminant analysis effect size (LEfSe) with an LDA score threshold of >3 [18]. Correlations between urea levels, environmental variables, differentially abundant fungi, and fungal community and functional composition were evaluated using the MANTEL test, which was conducted via the Genescloud tool (https://www.genescloud.cn, accessed on 20 December 2024) of Shanghai Personalbio Technology Co., Ltd. (Shanghai, China).

### 2.4. Isolation and Identification of Chaetomium sp. and Verification of Its Potential Function

The dilution coating method was employed to isolate and purify a strain of *Chaetomium* sp. from a rhizosphere soil sample of *L. barbarum*. To ascertain the taxonomic position of the isolated strain, its ribosomal DNA internal transcribed spacer (rDNA-ITS) gene region was Sanger sequenced. The primers ITS1 (TCCGTAGGTGAACCTGCGG) and ITS4 (TCCTCCGCTTATTGATATGC) were utilized for polymerase chain reaction (PCR) amplification. The ITS sequences of *Chaetomium* sp. were compared against those of known fungal species recorded in the NCBI database to identify closely related species. The phylogenetic tree of *Chaetomium* sp. was subsequently constructed using MEGA 11 software [21] to elucidate the evolutionary relationships among fungi, based on the maximum likelihood method. The solid-state fermentation of *Chaetomium* and the preparation of *Chaetomium* products (Powder, effective spore count 2 × 10^7^ cfu/g) were completed by Shandong Tianrunhe Bioengineering Co., Ltd. (Jinan, China).

To investigate the potential function of *Chaetomium* sp., pot experiments involving four nitrogen enrichment levels (N0, N1, N2, and N3) were conducted. Each nitrogen treatment included two treatments: CK (control, no addition of *Chaetomium* sp.) and LC101 (the fertilizer of *Chaetomium* was basally applied, with 1.5 g per plant). After 30 days of growth, the biomass, root viability, and chlorophyll content of the *L. barbarum* plants and the soil enzyme activities of the *L. barbarum* rhizosphere soil were measured. Chlorophyll was extracted from the leaf samples using ethanol, and the absorbance values at wavelengths of 645 nm and 663 nm were measured via a spectrophotometer to calculate the total chlorophyll concentration [22]. A TTC (tetrazolium chloride) reduction assay was used to evaluate the root vitality of *L. barbarum*, and the intensity of the color was measured spectroscopically at 485 nm [23]. The activities of soil nitrite reductase, soil nitrate reductase, soil polyphenol oxidase, and soil urease were measured via ELISA kits, and the tests were provided by Nanjing Ruiyuan Biological Technology Co., Ltd. (Nanjing, China) as a testing service.

### 2.5. Statistical Analysis

The results of the physiological analyses are presented as the means ± standard deviations of at least three biological replicates. Statistical analysis was performed via SPSS 22.0 software (IBM, Armonk, NY, USA). One-way ANOVA, along with the Student–Newman–Keuls test, was performed to determine statistically significant differences between the means of the groups.

## 3. Results

### 3.1. Effect of Nitrogen Supply Level on the Diversity of the Soil Fungal Community

The α diversity of the soil fungal community was assessed via multiple indices, including the observed species richness (Sobs), richness estimators (ACE and Chao1 indices), diversity indices (Shannon and Simpson), and community coverage (coverage). No significant differences in the α diversity indices of the soil fungal communities were detected between the nitrogen-treated samples (N1, N2, and N3) and the control sample (N0). Under nitrogen deficiency (N1) and excessive nitrogen (N3), the Shannon index was significantly lower than that under the control (N0) and normal nitrogen (N2) treatments. In contrast, compared with those in the control (N0) and normal nitrogen (N2) treatments, the Sobs, Simpson, ACE, and Chao1 indices were greater under nitrogen deficiency (N1) and excessive nitrogen (N3) (Figure 1A). Specifically, under normal nitrogen treatment (N2), the Sobs and Shannon indices were greater than those of the control (N0), whereas the Simpson index was lower. Spearman’s correlation analysis further demonstrated that the Sobs and Simpson indices were positively correlated with the nitrogen fertilizer application rates (Figure 1B).

Effect of nitrogen supply level on the beta diversity of the soil fungal community. The ordination analysis of the microbial community structure suggested that nitrogen addition levels may significantly affect the functions of the soil microbial community structure. PCoA revealed distinct clustering of microbial communities among groups (No, N1, N2, and N3) along the first two axes (Axis 1: 41.66%; Axis 2: 17.9%). The No group was separated from the N1 and N3 groups in the negative direction of Axis 1, whereas the N2 samples presented high intragroup dispersion. PERMANOVA confirmed significant differences between groups (R^2^ = 0.723, F = 6.706, *p* = 0.001), indicating that 72.3% of the variation was explained by group classification (Figure 2A). Redundancy analysis (RDA) revealed significant correlations between soil pH, electrical conductivity, organic matter content, total nitrogen, phosphorus, potassium, nitrate nitrogen, ammonium nitrogen, available potassium, available phosphorus, β–1,4-N-acetylglucosaminidase (NAG), the leucine aminopeptidase (LAP) urea level of urease (UR) and nitrogen fertilizer application levels (urea level), and the distribution of the fungal community in the rhizosphere of goji berries (Figure 2B and Table 1).

### 3.2. Effect of Nitrogen Supply Level on the Soil Fungal Community Composition

The ITS sequencing results revealed 11 fungal phyla in the rhizosphere soil of *L. barbarum*. In the experimental area, Ascomycota, Mortierellomycota, Blastocladiomycota, Basidiomycota, Chytridiomycota, Olpidiomycota, Rozelomycota, Zoopagomycota, Glomeromycota, Mucoromycota, Ascomycota (75.66%), and Mortierella (18.32%) were the dominant microorganisms, accounting for 93.95% of the fungal population abundance (Figure 3A). At the genus level, 370 fungal species were found, including Mortierella (17.97%), *Gibberella* (12.52%), *Fusarium* (10.01%), *Acaulium* (7.63%), Stachybotrys (2.32%), Alternaria (2.28%), *Lophotrichus* (2.01%), Dactylonectria (1.82%), *Chor-domyces* (1.80%), *Cladosporium* (1.80%), and Cephalotrichum (1.70%) (Figure 3B). At the genus level, compared with the N-treated samples (N1, N2, and N3), the control samples (N0) contained 160 fungi, among which Mortierella (18.56%), *Gibberella* (12.93%), *Fusarium* (10.34%), and *Acaulium* (7.88%) constituted the core fungal populations in the rhizosphere of Lycium. The LEfSe analysis revealed significant differences in microbial composition among the four groups (N0, N1, N2, and N3). Specifically, group N2 showed enrichment of features such as *Cladosporium*, and Chaetomum, group N3 was characterized by the abundance of *Gibberella*, and *Fusarium* was significantly enriched in the N1 treatment. In contrast, group N0 did not exhibit any prominently enriched features (Figure 3C).

### 3.3. Correlations Between Soil Environmental Factors and Fungal Community Composition Under Different Nitrogen Application Levels

We observed important correlations between soil chemical properties and soil fungal composition and diversity. The MANTEL test was used to analyze the relationships between soil chemical properties and fungal community OTU composition and fungal diversity under the four treatments (Figure 4A). The Mantel test results revealed a significant positive correlation between the composition of fungal communities and the urea level (R = 0.8226, *p* = 0.001), soil electrical conductivity (R = 0.5618, *p* = 0.004), organic matter (R = 0.3934, *p* = 0.005), total nitrogen (R = 0.4571, *p* = 0.006), nitrate nitrogen (R = 0.6671, *p* = 0.001), ammonium nitrogen (R = 0.4556, *p* = 0.005), and available phosphorus (R = 0.379, *p* = 0.007). Meanwhile, the fungal composition was significantly and positively correlated with important fungi (genus) (Figure 4B), such as *Stemphylium* (R = 0.2369, *p* = 0.048), *Mallochia* (R = 0.3345, *p* = 0.016), *Acaulium* (R = 0.8103, *p* = 0.001), and *Lophotrichus* (R = 0.7376, *p* = (0.001). A significant positive correlation was noted between *Lectera* (R = 0.5470, *p* = (0.003) and fungal alpha diversity. Although there was a positive correlation between the composition and diversity of *Chaetomium* and fungal community composition and diversity, the correlation was not significant.

### 3.4. Isolation and Identification of Chaetomium sp. and Verification of Its Potential Function

A *Chaetomium* isolate (LC101) was obtained from the rhizosphere soil of *L. barbarium* (Figure 5A). In the PDA medium, the growth pattern of the colonies was cloud-like, with dark brown mycelium, and the colonies also exhibited the ability to produce spores. Moreover, LC101 exhibited antagonistic activity against *Fusarium* oxysporum, which causes root rot of *L. barbarium*, in petri dishes (Figure 5A). Its accurate taxonomic status needs to be further confirmed using molecular techniques. *Chaetomium* sp. LC101 was characterized on the basis of ITS gene sequencing. Figure 5B illustrates the phylogenetic tree showing the evolutionary relationship of *Chaetomium* sp. LC101 with other closely related taxa. This isolate shared 92% similarity with *Chaetomium* globosum. Therefore, the isolate belongs to a species of *Chaetomium* globosum, which is also consistent with our microbial sequencing data mentioned above.

In the various nitrogen fertilizer addition treatments, the application of *Chaetomium* sp. strain LC101 significantly promoted the growth of *L. barbarum* (Figure 5C–G). In the N0 treatment (no additional nitrogen), the growth-promoting effect of *Chaetomium* sp. LC101 was the most significant. Compared with those of the control treatment, the fresh weight, root viability, and chlorophyll content of the *L. barbarum* seedlings increased by 55.10%, 15.69%, and 43.27%, respectively (Figure 5C,D). These enhancements suggest the beneficial role of *Chaetomium* sp. LC101 in improving plant health and nutrient uptake capacity under nitrogen-deficient conditions. Considering the combined effects of nitrogen fertilizer application rates and *Chaetomium* sp. strain LC101, under conventional nitrogen application (N2) conditions, inoculation with *Chaetomium* sp. LC101 resulted in the most significant increase in the growth of *L. barbarum* seedlings. The effectiveness of *Chaetomium* sp. LC101 in enhancing root fresh weight, root vitality, and chlorophyll content varied across different nitrogen fertilizer levels, in the following order: N0 > N1 > N2 > N3 (Figure 5E–G). These findings suggest that the synergy between nitrogen input and *Chaetomium* sp. LC101 inoculation optimally supports plant growth under moderate nitrogen conditions, whereas the growth-promoting effect was attenuated at relatively high nitrogen levels.

We further analyzed the impact of *Chaetomium* sp. LC101 on soil nitrogen transfer through changes in soil enzyme activities. In the various nitrogen fertilizer addition treatments, the addition of *Chaetomium* sp. LC101 significantly influenced the soil enzyme activity (Figure 6). The addition of *Chaetomium* sp. LC101 effectively increased the activity of urease (S-UE), nitrite reductase (S-NIR), and polyphenol oxidase (S-PPO) and inhibited nitrate reductase activity (S-NR) in the soil. Compared with that in the control treatment without *Chaetomium* sp. LC101, the activity of nitrite reductase in the N1 treatment with *Chaetomium* sp. LC101 increased by 55.48%; the activity of polyphenol oxidase in the N0 treatment increased by 45.51%; and the activity of urease in the N3 treatment increased by 65.26%. These findings indicate that there are nuanced interactions between fungi and the soil nitrogen cycle under different nitrogen regimes. Overall, these changes indicate the role of *Chaetomium* sp. LC101 in the balance of nitrogen transformation in the rhizosphere soil of *L. barbarum*.

## 4. Discussion

Nitrogen, a pivotal nutrient regulating terrestrial ecosystem productivity, exerts profound impacts on soil microbial community dynamics. Our study revealed that varying nitrogen input levels significantly altered the structure, diversity, and functional potential of the rhizosphere fungal community of *L. barbarum*, with nitrogen input intensity directly modulating fungal diversity and community stability. Consistent with previous research, our results demonstrated that high nitrogen deposition destabilizes fungal networks in semi-arid grasslands by favoring fast-growing eutrophic taxa over oligotrophic groups [24]. The reduction in the Shannon index under nitrogen-deficient (N1) and excessive (N3) conditions supports the nutrient imbalance hypothesis, which posits that both insufficient and excessive nitrogen availability disrupt microbial niche partitioning [25]. Distinct shifts were observed in the relative abundance of specific taxonomic groups, with a notable enrichment of *Chaetomium* under conventional nitrogen (N2) conditions. These findings not only align with broader ecological patterns but also provide novel insights into the interplay between nitrogen management, fungal functional traits, and plant–microbe interactions.

Soil microbes mediate soil nitrogen cycling, supply plants with nitrogen, and maintain soil fertility [26,27]. Fungi constitute an important component of rhizosphere microorganisms and are sensitive to changes in the soil microenvironment and fertilization level. Increased nitrogen input has led to soil acidification, reducing the soil pH and soil microbial diversity [2,28,29]. The long-term application of nitrogen fertilizer significantly reduces the Shannon index, chaos index, and fungal biodiversity and negatively impacts plant root fungi, such as arbuscular mycorrhizal fungi [10,11,30,31]. At the phylum level, Ascomycota and Mortierella were the dominant microorganisms. At the genus level, the N0 treatment enriched the *Acaulium* genus, the N1 treatment increased the Fusarium genus, and the N3 treatment enriched the *Gibberella* genus. The identification of fungal communities in the rhizosphere soil of *L. barbarum* revealed that there were significant differences in fungal species among the different urea treatments. The enzyme inhibition theory suggests that the application of a high concentration of exogenous nitrogen inhibits the activity of oxidases and reduces microbial carbon utilization, and that increasing nitrogen application can significantly affect the microbial community composition [32,33]. The low nutrient-symbiotic nutrition theory suggests that increasing nitrogen application can hinder the growth of hanging nutrient groups, whereas the oligotrophic symbiosis theory suggests a negative correlation between the *Fusarium* genus and the nitrogen application rate [34]. In addition, we confirmed that under high concentrations of urea, *Fusarium* in the soil is no longer the dominant fungus, which could be attributed to the different responses of oligotrophic and copiotrophic groups to different resource utilizations [35]. Only a few studies have reported a correlation between the nitrogen application rate and soil fungi. Microorganisms in the soil interact with environmental factors and complement each other, indicating the importance of conducting further studies.

Soil fungi play a pivotal role in maintaining the delicate balance of nitrogen (N) in the soil; however, this ability is rarer among fungi and contributes to the overall nitrogen stock in soil ecosystems. During periods of high nitrogen availability, some fungi can immobilize nitrogen, storing it within their biomass, such as Suillus, Laccaria, and Hebeloma [36,37]. When conditions become nitrogen-limited, this stored nitrogen can then be released back into the soil, serving as a buffer mechanism that helps maintain nitrogen availability over time. Fungi, therefore, through their enzymatic capabilities, can control the flow of nitrogen in ecosystems by regulating the balance between nitrogen retention and loss, such as nitrification and denitrification [38]. Their impact extends beyond direct nitrogen transformation, as the activities of these enzymes influence the overall microbial community structure and the efficiency of nutrient cycling. Studies have shown that fungi can dominate these enzymatic processes under certain conditions, making them key regulators of soil nitrogen dynamics [5]. In our study, the application of varying amounts of nitrogen fertilizer significantly affected the composition of the fungal communities found in the rhizosphere soil of *L. barbarum*. Among the changes observed, there was a noticeable increase in the presence of fungi belonging to *Chaetomium*, which may play an important role in the balance of nitrogen transformation in the rhizosphere soil of *L. barbarum*. Increased urease and nitrite reductase activities may increase the availability of ammonium nitrogen and support plant growth, whereas decreased nitrate reductase activity might initially slow the conversion of nitrate to ammonium and may require adjustment of management measures to ensure optimal nitrogen supply to plants.

## 5. Conclusions

The application of different nitrogen sources to the soil at varying levels significantly altered the soil microbial community structure. The urea application level significantly correlated with the fungal community structure composition, functional composition, and relative abundance of *Purpureocillium*, *Gibberella*, *Acaulium*, *Lophotrichus*, *Dactylectria*, *Fusicolla*, *Chaetomium*, and *Scutellinia*. The composition and function of fungal communities were significantly and positively correlated with the organic matter content and ammonium nitrogen content. In addition, *Chaetomium* sp. LC101 was obtained from the rhizosphere soil of *L. barbarum*. *Chaetomium* sp. LC101 may play an important role in promoting plant growth and the balance of nitrogen transformation. This finding highlights the sensitivity of soil fungal communities to exogenous nitrogen inputs such as nitrogen fertilizers. Understanding these effects is crucial for developing sustainable farming practices that not only increase crop yield but also maintain soil quality in the long term. Future studies should focus on optimizing nitrogen management strategies to promote beneficial fungal populations while minimizing the potential negative consequences of nitrogen application on soil biodiversity and function.

## Figures and Tables

**Figure 1 microorganisms-13-01864-f001:**
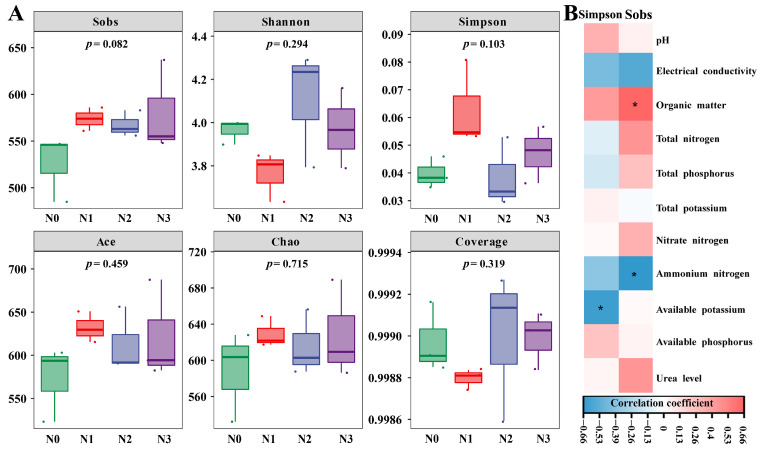
Effect of nitrogen addition on fungal alpha diversity indices in the *L. barbarum* rhizosphere. (**A**) Diversity alpha index; (**B**) Spearman correlation between the diversity alpha index and environmental factors. Note: The jitters correspond to the values from each of the three repeats. Note: * indicate statistically significant correlation between the samples.

**Figure 2 microorganisms-13-01864-f002:**
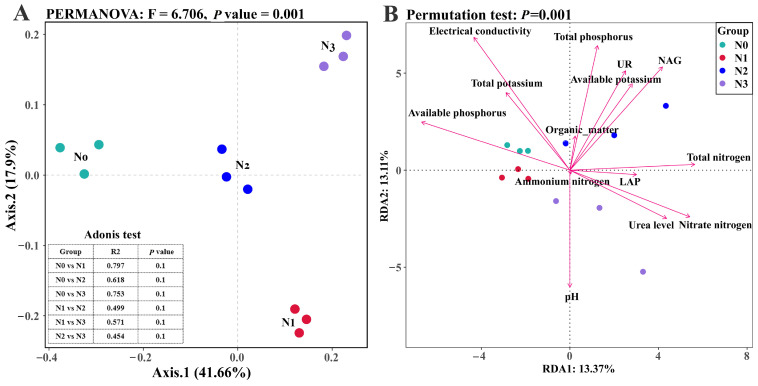
Effect of nitrogen addition on fungal ordination analysis in *L. barbarum* rhizosphere. (**A**) Unconstrained principal coordinate analysis (PCoA); (**B**) redundancy analysis (RDA).

**Figure 3 microorganisms-13-01864-f003:**
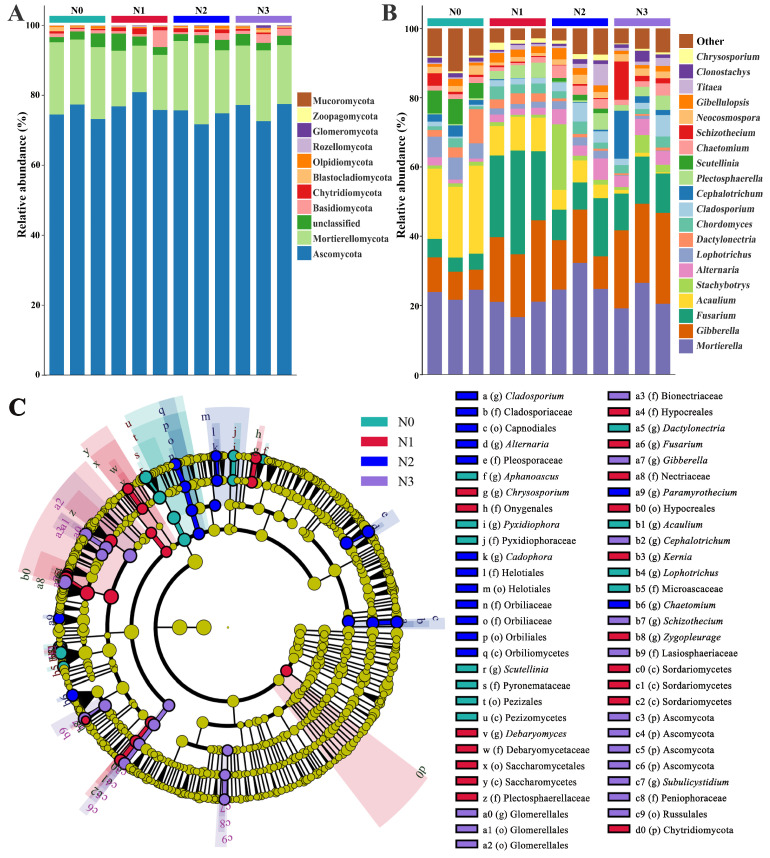
Effect of nitrogen addition on the fungal community in *L. barbarum* rhizosphere. (**A**) Phylum-level composition; (**B**) Genus-level composition; (**C**) Lefse multi-level discriminant analysis.

**Figure 4 microorganisms-13-01864-f004:**
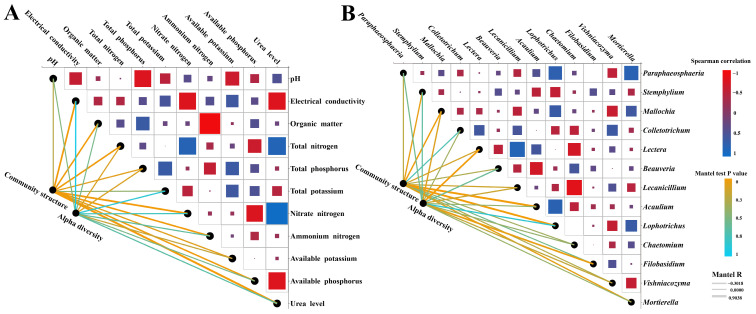
Mantel correlation analysis between rhizospheric microorganism *of L. barbarum* and soil environmental factors under different nitrogen application levels. (**A**) Correlation analysis between the microbial community and soil environmental factors. (**B**) Correlation analysis between different fungi in the rhizosphere of *L. barbarum*.

**Figure 5 microorganisms-13-01864-f005:**
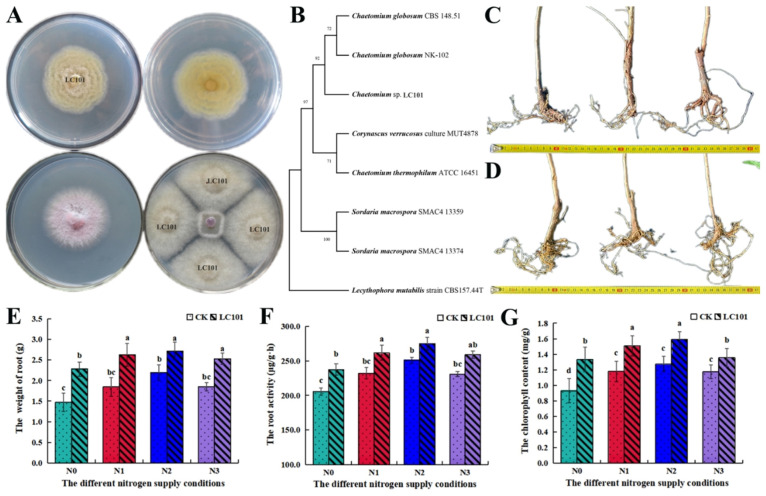
Isolation and characterization of *Chaetomium* sp. LC101 from the rhizosphere soil of *L*. *barbarum* and its growth-promoting effect on *L. barbarum*. (**A**) Colony morphology of strain LC101 and its antagonistic activity against *Fusarium oxysporum*; (**B**) phylogenetic tree of strain LC101 based on sequence analysis; (**C**) root system of *L. barbarum* in the control group (without strain LC101 inoculation); (**D**) root system of *L. barbarum* in the treatment group (inoculated with strain LC101); (**E**) effect of strain LC101 on the fresh weight of *L. barbarum* root tissues in different treatment groups; (**F**) effect of strain LC101 on the activity of *L. barbarum* in different treatment groups; (**G**) effect of strain LC101 on the chlorophyll content in *L. barbarum* leaves in different treatment groups. Note: Different letters above the bars indicate statistically significant differences between the samples (one-way ANOVA followed by a post hoc Tukey test, *p* < 0.05).

**Figure 6 microorganisms-13-01864-f006:**
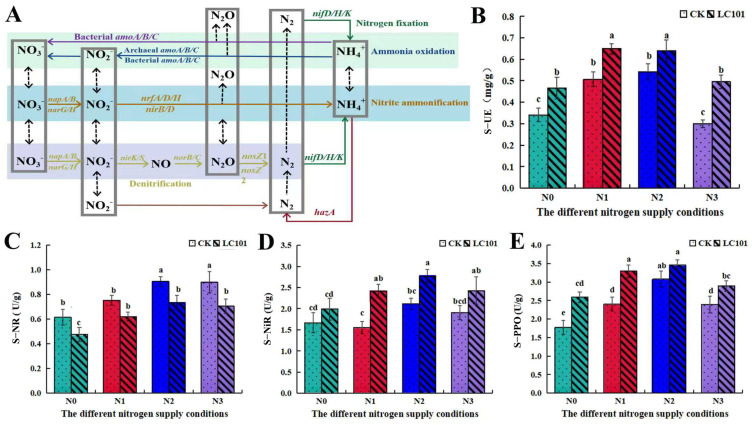
The role of *Chaetomium* sp. LC101 in the balance of nitrogen transformation. (**A**) Schematic view of the nitrogen cycle in soil and the key genes involved (from [1]). (**B**) Activity of urease; (**C**) activity of nitrate reductase; (**D**) activity of nitrite reductase; (**D**) activity of polyphenol oxidase; (**E**) activity of nitrate reductase. Note: Different letters above the bars indicate statistically significant differences between the samples (one-way ANOVA followed by a post hoc Tukey test, *p* < 0.05).

**Table 1 microorganisms-13-01864-t001:** Correlation analysis of environmental factors and fungal community structure variation based on redundancy analysis (RDA) determined by a forward selection procedure with unrestricted permutation tests.

Environmental Factors	Redundancy Analysis			
	RDA1	RDA2	R^2^	*p* Value
pH	−0.0001	−1.0000	0.4616	0.0210 *
Electrical conductivity	−0.5348	0.8450	0.6205	0.0060 **
Organic matter	0.1351	0.9908	0.1363	0.6307
Total nitrogen (N)	0.9986	0.0526	0.4322	0.0275 *
Total phosphorus (P)	0.1908	0.9816	0.5009	0.0090 **
Total potassium (K)	−0.5846	0.8114	0.3779	0.1239
Nitrate nitrogen	0.9137	−0.4064	0.4537	0.0275 *
Ammonium nitrogen	0.1904	−0.9817	0.0160	0.9440
Available potassium	0.5302	0.8479	0.4203	0.0405 *
Available phosphorus	−0.9377	0.3475	0.5470	0.0025 **
NAG	0.6175	0.7865	0.5182	0.0285 *
LAP	0.9971	−0.0759	0.2310	0.3323
UR	0.4403	0.8979	0.4372	0.0550
Urea level	0.8684	−0.4958	0.3850	0.0935

Note: The soil chemical properties and soil enzyme activity were obtained from our previous research [1]. * indicate the soil chemical properties or soil enzyme activity have a significant impact on the fungal community structure at the 0.05 level. ** indicate the soil chemical properties or soil enzyme activity have a significant impact on the fungal community structure at the 0.01 level.

## Data Availability

The data in Table 1 comes from published data by our team in this study. The Sequencing data presented in this study are openly available in NCBI at [https://dataview.ncbi.nlm.nih.gov/object/PRJNA1274136?reviewer=oaarc36hhq1srpdc574c2buuqr], reference number [PRJNA1274136].

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
