# Peer review of "Response of Chaetomium sp. to Nitrogen Input and Its Potential Role in Rhizosphere Enrichment of Lycium barbarum"

_microorganisms, 2025, doi:10.3390/microorganisms13081864_

Round 1
Reviewer 1 Report (Previous Reviewer 2)
Comments and Suggestions for Authors
The manuscript, titled "Response of Chaetomium sp. to nitrogen input and its potential role in rhizosphere enrichment of Lycium barbarum", presents the results of a study aimed at improving the cultivation of a plant (goji berry) long used in traditional Chinese medicine. Numerous papers have been published on the role of bacteria in nitrogen fixation processes in the soil, but there is relatively little knowledge about nitrogen cycling mediated by fungi. The authors continued their studies on this topic by establishing several nitrogen supply levels (0, deficiency, normal supply and overdose). In addition, very intensive identification work was carried out using molecular biology methods to identify fungi that could be isolated from the soil, and correlations were established regarding the effects of soil environmental parameters on the species composition of fungal communities.The positive effects of a soil-dwelling fungus species Chaetomium isolated from the root zone of goji berry on the root growth and other plant physiological parameters of Lycium barbarum were also investigated.
The introductory part is sufficiently detailed and well presents the related research on the topic to date and its results.
The materials and methods chapter also presents the experimental site, the applied test methods and materials as well as the statistical analysis procedures in sufficient detail.
The presentation of the results is, however, well-structured, sufficiently detailed, and illustrated with spectacular figures.
The conclusions are formulated in a moderate manner. The results are compared with similar results published in international literature sources. Number of cited references are: 36.
I recommend publishing the manuscript as a scientific article.
Author Response
Dear Reviewer:
We would like to thank you for your comments you have made on our manuscript. (Manuscript ID: 3764388).
Reviewer 2 Report (Previous Reviewer 1)
Comments and Suggestions for Authors
Abstract
Line 18. Please correct the first sentence, it’s incomplete. Please substitute “chemical” with “chemically-synthesized”
Line 19. Please correct “remains understudied”. First, there are thousands of studies demonstrating the impact of chemically-synthesized agrochemicals in fungal communities, and second, the appeal to the “ad ignorantium” logical fallacy to justify a scientific work is unacceptable.
Line 24. Please substitute “gradients” with “conditions”. Please delete “three nitrogen states of”
Line 26. Please do not subscript the number 1 in ITS1
Lines 30-31. Do not use “low”, “high,” and “normal” to define a nitrogen concentration; instead, please state the actual concentration.
Line 34. Please consolidate the narrative for Chaetomium isolation in a single mention, fusing with the same description appearing in line 25
Lines 30-42. Please rewrite this section; it uses a dogmatic discourse instead of a cause-and-effect scientific discourse.
Keywords
Line 41. Please do not use words or terms already appearing in the title
Introduction
Line 52. Please substitute “excessive” with “high”.
Line 60. Please substitute “deficient” with “deficiency”.
Line 63. Please add a space between “potential” and “function”.
Line 66. Please italicize Glomus
Line 79. Please, in the first mention of Lycium barbarum, mention the full scientific name as follows: “Lycium barbarum L. (Solanaceae)”
Materials and methods
Line 98. Please substitute N2 as a treatment name, as it may be confusing with the formula of molecular nitrogen, N2. I suggest using “L-N” for low nitrogen, “M-N” for medium nitrogen, and “H-N” for high nitrogen dosage. Please correct this terminology in all sections and figures of the document. Use “Ctrl” for no nitrogen addition.
Line 100. Please delete “These treatments characterize nitrogen deficiency (N1), normal (N2), and nitrogen excess (N3), respectively.”
Lines126-128. Please do not subscript 1 and 2 in ITS1–ITS2.
Line 140. Please state the exact number of samples that were sequenced. Please present as a supplementary file the “in-house Perl script”.
Line 145. Please explain why the term “16S rRNA” is mentioned. This is only evidence of copy-paste bad writing practices.
Line 149. Please explain “significant differences” of what?
Line 158. Please explain what “spatial compartments” refers to.
Line 165. Please use “L. barbarum” instead of “Gogi berry”.
Lines167-169. Please do not subscript 1 and 4 in ITS1 and ITS4.
Line 178. Please state clearly the exact formula of the Chaetomium product, including the CFU per volume or weight, adjuvants, etc.
Line 180. In lines 97 and 98, only three nitrogen levels were mentioned, but here, four of them are mentioned. Please correct this inconsistency and eliminate it from this section to avoid repeated and confusing information.
Line 182. Please state the exact Chaetomium sp. dosage applied and the mode of application.
Line 185. Please add a space before “nm”
Line 190. Please state the catalog number of the ELISA kits. Apparently, this company doesn’t sell these products https://www.pronetbio.com/products_list_3/34.html
Results
Line 199. Please include here a new section describing the sequence report, including the public accession to raw sequences, the total number of bp obtained, the average number of reads per sample, and the sequence quality of reads on the Phred scale.
Line 215. Figure 1. Please explain what the jitters in the graphs from panel A correspond to. Are these the values from each of the three repeats?
Line 244. Table 1. Please remove the Nitrogen treatments columns, as this was already published. Leave only values for redundancy analysis.
Line 268. Figure 2. For consistency and clarity, please use in panel C the same color code for treatments employed in the previous figures.
Line 291. Figure 4. Please remove the “beta diversity” in panels A and B. No beta diversity was calculated.
Line 295. Please remove “and soil environmental factors”.
Line 298. Please correct to L. barbarium.
Line 305. Please italicize Fusarium oxysporum
Line 314. Figure 5. Please use the same color code employed in previous figures in graphs from panels E-G. Please eliminate the word “The” from the x- and y-axes labels in panels E-G
Line 327. Please italicize Chaetomium sp.
Line 360. Figure 6. Please use the same color code employed in previous figures in graphs from panels B-E. Please eliminate the word “The” from the x-axis labels in panels B-E.
Discussion
This section is poorly written. A strong review is needed to revise the previous reports on the ecological and physiological roles of the taxonomic groups showing shifts in relative abundance, with particular interest in Nitrogen metabolism. The internal and external discussions do not explore the putative role of the added Chaetomium on the microbial community.
Lines 376-380. Please delete “We found… urea level.” These are results previously published elsewhere.
Lines 286-387. Please substitute “species” with “genus”.
Line 396. Please italicize “Fusarium”, please change “species” with “genus”.
Line 398. Please italicize “Fusarium”.
Line 405. Please italicize “Suillus”, “Laccaria”, and “Hebeloma”.
Comments on the Quality of English Language
The manuscript requires a professional grammar and scientific style review.
Round 2
Reviewer 2 Report (Previous Reviewer 1)
Comments and Suggestions for Authors
Thank you for attending the comments.
This manuscript is a resubmission of an earlier submission. The following is a list of the peer review reports and author responses from that submission.
Round 1
Reviewer 1 Report
Comments and Suggestions for Authors
General comments:
Please correct the citation style following the Journal’s instructions for authors. Please use the Journal’s format for section and subsection levels.
Title
Line 1. Please add “(Ascomycota)” after “Chaetomium sp.”
Line 2. Please add “(Solanaceae)” after “barbarum”
Abstract
Line 14. Please remove “relies heavily on nitrogen for growth”. Every living being does.
Line 20. Please use “L. barbarum”
Line 21. Please insert “ITS amplicon” after “high-throughput”
Line 22. Please insert “Sanger” after “ITS1”. Please remove “analysis”
Line 23. Please remove the letter “R” before “The structure”
Lines 25-26. The terms “low”, “high”, and “normal” are ambiguous; please refer to a "N" concentration in soil.
Line 30. Please substitute “significantly contributed” with “may contribute”. The authors cannot discard this effect, which was caused by another microbial component.
Line 31. Please change “can” with “may”. Please avoid dogmatic discourse.
Line 33. Please change “can” with “may”.
Line 33. Please change “will” with “would”.
Line 38. Please change "healthy” to “promoting a healthier”
Keywords
Line 40. Please remove any word already appearing on the Title
Introduction
The introduction must be rewritten to incorporate previously generated evidence on the effect of nitrogen addition on soil microbial communities, with special detail on fungal communities. The introduction must incorporate a hypothesis-guided discourse and justification instead of a descriptive revision of the literature. In the introduction
Line 43. Please remove “element”. Please substitute “the major element” with “a macronutrient”. After the first mention of nitrogen, please add “(N)” and use N for fthe ollowing mentions.
Line 44. Please replace “Life activities” with “life on earth” and remove " the majority of the”. All plants do.
Line 46. Please describe the “side-dressing” cultural practice. This is not an agronomy journal, so the audience may not be familiar with this terminology.
Line 52. Please remove “However,”
Line 59. Please remove “such that”
Line 61. For clarity, please refer to the specific reaction involving “nitrogen fixation, nitrification, denitrification, and ammonification”.
Line 62. Please substitute “Research” with “Previous evidence”
Line 63. Please correct “rhizobia” to a valid taxonomic clade.
Line 64. For clarity, please refer to the specific reaction involving “nitrification”.
Line 70. Please change “can effectively reflect the” with “may provide insights on”.
Line 72. Please use “effect” in singular
Line 73. Please remove “microorganisms”. Please substitute “thereby improving” with “putatively involved on”
Line 74. Please substitute “main 7-year-old” with “L. barbarum”
Line 75. Please remove ““Ningqi No.7””. Please substitute “object” with “model”
Line 76. Please change “exhibit” to “evaluate”
Line 77. Please remove “i.e., nitrogen deficiency, normal, and nitrogen excess”
Materials and Methods
Line 84. Please use “germplasm” in singular
Line 85. Please correct to “nursery”. Please remove “of Lycium barbarum”
Line 86. Please remove “~”. Please use “masl” instead of “m alt.”
Line 88. Please remove “hours”. Please replace “Blank” with “No N added. Please state here the employed fertilizer as follows: Commercial name (Catalog number, manufacturer, city and country of manufacturer’s headquarters). Then, please provide detailed purity and formulation of the fertilizer in terms of N form or forms present in the product, as well as N weight per gram of product. Finally, express N addition in g·of product added m–2·year–1, distributed on N number of applications on months X, Y, Z.
Line 90. Please describe the experimental design.
Line 91. Please substitute “67.62 g·N m–2·year–1” with “N2”.
Lines 92-93. Please remove “These treatments… …respectively.”
Line 93. Please remove “The main”. Please change “7” with “Seven”. Please insert “plants from the” after “L. barbarum”
Line 94. Please change “was” with “were”
Line 95. “single basal application” of what? Please clarify.
Lines 96-97. Please substitute dates with days after sowing.
Lines 97-99. Please remove “Gogi berry… … maize”
Line 100. Please substitute dates with days after sowing.
Line 106. Please insert “amplicon” and “experiments” before and after “sequencing” respectively.
Line 109. Please substitute “our” with “a”
Line 110. Table S1. Please remove it. Previously published data cannot be published again; it consists of self-plagiarism.
Line 122. The description of library preparation and sequencing protocol followed is missing. Please include.
Line 126. Please cite the “PEAR software”. Please remove “which is a fast and accurate Illumina paired-end reAd mergeR”
Line 127. Please remove “The maximum mismatch rate in the overlapping areas was constrained to 0.1.”
Line 128. Please cite the “UCHIME algorithm”
Line 131. Please cite “BLAST”. Please cite the “UNITE database” and state the version used.
Line 134. Please cite the “Mothur” and “Vegan” packages.
Line 138. Please cite “R”. Please remove “package”
Line 139-140. Please substitute “evaluate the beta diversity of” with “ordain data from” Ordination analysis are not beta diversity indices. Beta diversity indices are Whittaker, Harrison, Cody, Routledge, Wilson-Shmida, Mourelle: Harrison 2, or Williams.
Line 144. Please explain what “compartment-specific” refers to. Please use consistent terminology.
Line 145. Please explain what “biomarkers” refers to, and please explain how they were identified.
Line 147. Please capitalize “MANTEL”
Line 148. Please clarify why MANTEL correlations were performed against the urea level only instead of all other N forms evaluated in Li et al., 2023.
Line 149. Please capitalize “MANTEL”
Line 150. Please remove “which is a free online platform” and “for data analysis developed by”
Lines 153-154. Please describe in detail the bacterial isolation method, including sampling, media used, incubation, subcultures, and selection criteria (morphologic, phenotypic, etc.) Please describe the method employed for DNA extraction.
Line 156. Please insert “Sanger-“ before “sequenced”
Line 158. Please delete “of the internal transcribed spacer (ITS) region specific to fungi”
Line 161. Please cite the MEGA software
Line 162-164. Please describe in detail the procedures for “solid-state fermentation of Chaetomium and preparing Chaetomium products”. The methodology sections MUST include ALL the details to lead other researchers to repeat your research.
Line 165. Please italicize “Chaetomium”
Lines 166-167. Please clarify if those nitrogen treatments were the same as described in lines 88-90. If so, please mention only “N0-N3 treatments mentioned in section 2.1.
Lines 168-169. Please do not use internal coding of treatments (CK and LC101). These codes are confusing and counterintuitive. Please consider using “-Ch” and “+Ch” instead. Please state the dosage use of “Chaetomium sp” in cfu/mL, adding information on how the inoculum was prepared and applied to the pots, as well as other missing culture details such as time, watering conditions, density, experimental design, etc.
Line 168. Please italicize “Chaetomium”
Line 169. Please italicize “Chaetomium”
Line 170. Please italicize “L. barbarum”
Line 171. Please italicize “L. barbarum”
Line 171. Please describe the chlorophyll extraction method in detail.
Line 174. For all reactants, materials, commercial software, and instruments employed, please cite as “name (Catalog number, manufacturer (or developer), city and country of manufacturer’s (or developer’s) headquarters)”
Line 175. Please italicize “L. barbarum”
Line 176. Please delete “to estimate the degree of root vitality”
Line 178. Please delete “enzyme-linked immunosorbent assay,”. For all reactants, materials, commercial software, and instruments employed, please cite as “name (Catalog number, manufacturer (or developer), city and country of manufacturer’s (or developer’s) headquarters)”
Line 181. Please add a space between “±” and “standard”
Line 182. For all reactants, materials, commercial software, and instruments employed, please cite as “name (Catalog number, manufacturer (or developer), city and country of manufacturer’s (or developer’s) headquarters)”
Results
Line 205. Figure 1. Panel A. Please order the treatments from N0 to N3 in all graphs in panel A. Please include significance letters for treatments in each graph, adding the p value by pair of treatments (Please clarify to what comparison corresponds to the p value shown in graphs. Please use capital letters in the graph’s headings in panel A (Sobs, Shannon, Sipson, etc.). Please do not use bold letters in graphs. Panel B. Please increase the font size to fit the size in panel A, or at least 9 points on the screen. Please indicate in the figure what the color scale refers to.
Line 206. Please substitute “index” with “indices”
Line 207. Please use “alpha” after “diversity”
Line 208. Please indicate what the asterisks in panel B refer to.
Line 210. Please substitute “beta-diversity” with “ordination”. Ordination analyses are NOT beta diversity indices. Whittaker, Harrison, Cody, Routledge, Wilson-Shmida, Mourelle: Harrison 2, or Williams are some of them. Please calculate at least one of them if beta diversity is mentioned. Please substitute “revealed” with “suggest”
Line 211. Please substitute “significantly caused” with “may cause significant”
Line 224. Figure 2. Please show the significance of the p value by paired treatment comparisons in panel A. Please delete the current panel headings in both panels. Please do not use bold letters in both panels. Please use the same font size on both panels. Please change “Group” per “Treatment” in the panel B legend.
Line 225. Please substitute “beta-diversity index” with “ordination analysis”. Please italicize “Lycium barbarum”
Line 227. Please delete Table 1. Correlation must be performed by comparing taxa abundance vs. values from environmental values. Correlations comparing RDA are weak.
Line 230. Please move here the section to line 186 as the first results section and rename as “Structure of soil fungal community under different Nitrogen addition levels”. Before the current text, please mention here the sequencing results, station the exact number of files obtained by treatment and the public link to a free access repository like the NCBI. Please mention the total number of reads obtained, global read quality, and the average number of bp per sample. Please provide as supplementary material an Excel spreadsheet showing the absolute abundance of clades found at different taxonomic levels (From the phylum to species level).
Line 232. Please remove “L.” after “barbarum”
Line 232-235. Please do not italicize the phyla names.
Line 249. Figure 3. Please change “Sequence Number Percent(%)” with “Relative abundance (%)” in panels A and B. Please do not use bold letters in panels A-D. Please use the same font size in panels A-D. Please do not italicize “Other” in the legend of panel B. The letters in panel C are tough to read; please use a paler tone of colors, use grey lines, and an Arial font to make those letters easier to read. In the legend of panel C, please use “a (C) Pezizomycetes” (please substitute “:” with a space; please use the taxa level indicator as an uppercased letter between forceps, please substitute “_” with a space). Please italicize the genera-level names, such as “Scutellinia”. Please report only the taxa level where a significant identification has been reached (please change “d1:f_unclassified_o_Glomerallales” with “d1 (O) Glomerallales”. Panel D. Please substitute the current panel with a heatmap at the genus level with a two-way clustering analysis based on Bray-Curts distances.
Line 257. Why “bacterial” is mentioned here? Please clarify. This was an uncorrected copy-paste?
Line 261. Why “bacterial” is mentioned here? Please clarify. This was an uncorrected copy-paste?
Line 265 “functional composition of bacteria”???? What does this mean? This is evidence of a plagiarism practice. The text here was copy-pasted from another work without ant review or adaptation criteria. Moreover, no functional analysis was performed and no results regarding this matter are presented.
Line 283. Figure 4. Panel A should be removed as those data were partially published elsewhere (https://www.frontiersin.org/files/Articles/1070817/fmicb-13-1070817-HTML/image_m/fmicb-13-1070817-g001.jpg). Moreover, again, the authors mention “Bacterial community composition”, “Bacterial function composition”, “Fungal function composition” and “Bacterial α diversity” in this panel, but no experimental evidence here presented support this.
As a reviewer, I cannot continue with the review process at this point. The inconsistence findings goes beyond errors, they fall in a intentional unethical and careless paper preparation process. I strongly recommend rejecting and communicating this situation to PubPeer, Better Science, and Retraction Watch.
Comments on the Quality of English LanguageA professional service of scientific writing in English is recommended.
Reviewer 2 Report
Comments and Suggestions for Authors
The manuscript, titled "Response of Chaetomium sp. to nitrogen input and its potential role in rhizosphere enrichment of Lycium barbarum", presents the results of a study aimed at improving the cultivation of a plant (goji berry) long used in traditional Chinese medicine. Numerous papers have been published on the role of bacteria in nitrogen fixation processes in the soil, but there is relatively little knowledge about nitrogen cycling mediated by fungi. The authors continued their studies on this topic by establishing several nitrogen supply levels (deficiency, normal supply and overdose). In addition, very intensive identification work was carried out using molecular biology methods to identify fungi that could be isolated from the soil, and correlations were established regarding the effects of soil environmental parameters on the species composition of fungal communities.The positive effects of a soil-dwelling fungus species Chaetomium isolated from the root zone of goji berry on the root growth and other plant physiological parameters of Lycium barbarum were also investigated.
The introductory part is sufficiently detailed and well presents the related research on the topic to date and its results.
The materials and methods chapter also presents the experimental site, the applied test methods and materials as well as the statistical analysis procedures in sufficient detail. In the case of the table marked S1, however, it would be necessary to include a legend for the symbols in the treatment column (OM, TN,TP, TK, AK, AP, NAG, LAP, UR). The same applies to Table 1 in the presentation of the results chapter.
The presentation of the results is, however, well-structured, sufficiently detailed, and illustrated with spectacular figures.
The conclusions are formulated in a moderate manner. The results are compared with similar results published in international literature sources. Number of cited references are: 43.
After implementing the suggested additions, I recommend publishing the manuscript as a scientific article.